# How Do Different Dietary Carbohydrate/Lipid Ratios Influence Intestinal Morphology and Glycolipid Metabolism Capacity in Hybrid Grouper (*Epinephelus fuscoguttatus* ♀ × *E. lanceolatus* ♂)

**Xuanyi Yang** [1], **Xinwei Guo** [1], **Xiaohui Dong** [1,2], **Qihui Yang** [1,2], **Hongyu Liu** [1,2], **Shuang Zhang** [1,2], **Beiping Tan** [1,2,*] **and Shuyan Chi** [1,2,*] 

[1]  Laboratory of Aquatic Nutrition and Feed, College of Fisheries, Guangdong Ocean University, Zhanjiang 524088, China; yangxuanyi96@163.com (X.Y.); 15889833262@126.com (X.G.); dongxiaohui2003@163.com (X.D.); qihuiyang03@163.com (Q.Y.); liu_hong_yu@163.com (H.L.); zhangshuang198610@126.com (S.Z.)
[2]  Southern Marine Science and Engineering Guangdong Laboratory (Zhanjiang), Zhanjiang 524025, China
*  Correspondence: bptan@126.com (B.T.); chishuyan77@163.com (S.C.)

**Abstract:** Levels of dietary carbohydrates and lipids have important effects on the growth, health, and glycolipid metabolizing capacity of the hybrid grouper (*Epinephelus fuscoguttatus* ♀ × *E. lanceolatus* ♂). This experiment evaluated the effect of carbohydrate to lipid ratios (CHO:L ratio) on growth, serum biochemical indices, intestinal morphology and activity of digestive enzymes, and the ability to metabolize carbohydrates and lipids in the hybrid grouper. Six iso-nitrogenous (500 g/kg) and iso-energetic (190 MJ/kg) feeds with CHO:L ratios of 0.82, 1.03, 1.28, 1.58, 1.94, and 2.27 were formulated. The 720 groupers with an initial body weight of $7.76 \pm 0.08$ g were sorted into 24 fiberglass buckets of 30 fish each. Feed was delivered twice daily at 8:00 and 16:00 to apparent satiety for 56 days. The results showed that the growth performance of the groupers was not significantly affected by the diet. Based on the specific growth rate (SGR), the optimal CHO:L ratio was 1.72 for the groupers by the second-order polynomial regression model. The intestinal trypsin and lipase, and the hepatic trypsin activities of the groupers showed a maximum in the 1.58 group. The intestinal muscle thickness in the 1.94 group and the villus length in the 1.58 group were significantly greater than in the 0.82, 1.03, 1.28, and 2.27 groups. The glycogen synthase, pyruvate kinase, and hexokinase activities of the liver were significantly higher in the 1.58 group than in the 0.82 and 2.27 groups. The hormone-sensitive triglyceride lipase, adipose triglyceride lipase, acetyl-CoA carboxylase, and carnitine palmitoyl transferase of the liver showed significantly higher activities in the 1.58 group than those in the 0.82 and 1.03 groups. These results showed that based on the SGR of the groupers, the best CHO:L ratio was 1.72. An appropriate dietary CHO:L ratio can reduce the lipid deposition of the fish body, liver, and muscle, as well as improve the intestinal morphology and activities of liver carbohydrate and lipid metabolism enzymes.

**Keywords:** growth performance; carbohydrate to lipid ratios; intestinal morphology; carbohydrate and lipid metabolism; grouper

**Key Contribution:** The optimal ratio of carbohydrates/lipids in the grouper diet was determined, which provides a theoretical basis for the rapid development of the grouper industry.

## 1. Introduction

Protein is the most expensive nutrient that is essential for the growth and composition of the body [1,2]. The optimal feed is to reduce the degradation of protein, which is an energy source, improve the proportion of non-protein energy in the body, without affecting

the healthy growth of the fish, and maximize the protein utilization of the diet [3]. Carnivorous fish have a higher demand for dietary protein due to their carnivorous characteristics. It is widely acknowledged that the digestive and metabolic systems of carnivorous fish are better suited to using proteins and lipids rather than carbohydrates to provide energy [4,5], due to their glucose intolerance [6,7]. In order to use dietary protein effectively and reduce protein consumption for energy, it is necessary to increase the proportion of non-protein energy, such as carbohydrates and lipids, in the body's energy supply [3]. Carbohydrates and lipids not only serve as two important sources of energy but also regulate the growth performance and metabolism of fish [8]. Even so, the ability of different fish to use carbohydrates is also different; for example, the gilthead sea bream (*Sparus aurata* L.) and silver sea bream (*Sparus sarba*) can ingest high levels of carbohydrates as an energy source [9,10]. Most experiments have shown that the optimum levels of dietary carbohydrates can improve feed efficiency and growth in marine fish, for example, golden pompano (*Trachynus ovatus*) [5] and cobia (*Rachycentron canadum* L.) [11], with better nutrient absorption capacity; nevertheless, excess carbohydrates in the feed can lead to the poor growth of organisms, a lower lipid content in the whole body, a glucose burden [12,13], oxidative stress, suppressed innate immunity, and reduced disease resistance, thus affecting the health status [14–16].

Dietary lipid, playing a key role in fish nutrition, works as a source of energy and essential fatty acids [17]. However, higher dietary lipid levels easily result in an imbalance in the digestible energy/crude protein ratio and higher lipid deposition in tissues [18,19]. An optimal dietary lipid concentration is conducive to maximum growth [20,21]. The carbohydrates to lipids ratios (CHO:L ratios) in the feed significantly affect growth and digestion, as well as nutrient absorption, in aquatic animals. Numerous studies have shown that moderate CHO:L ratios can save protein [22], and improve feed utilization [23,24], growth performance [25–27], and non-specific immunity [28,29] of fish. Exploring the optimal CHO:L ratio of aquatic animal feeds could gain great instructive economic value to the development of the aquaculture industry [30].

In 2021, the total economic output value of the Chinese marine aquaculture industry reached 33.87 million; grouper production, which accounts for 11.07% of the total marine fish farming production, ranked third at 204,119 tons [31]. Hybrid grouper (*Epinephelus fuscoguttatus* ♀× *E. lanceolatus* ♂), with good properties of rapid growth, high disease resistance, and a high nutrition value, is farmed as a commercially important fish in the southern coastal areas of China [32,33]. The nutrient requirements of hybrid grouper, such as the amino acid profile [32], protein level [34], and protein to energy ratio [35], have been described in previous research. However, there are few studies on CHO:L ratios in the hybrid grouper. Therefore, this study evaluated the effects of different non-protein energy ratios on growth, whole body and tissue composition, morphometric indices, serum biochemical indices, and the digestive and glucolipid metabolism capacity of grouper to determine the optimal dietary CHO:L ratio for the hybrid grouper.

## 2. Materials and Methods

### 2.1. Experimental Design and Diets

As shown in Table 1, six iso-nitrogenous (500 g/kg) and iso-energetic (190 MJ/kg) feeds with CHO:L ratios of 0.82, 1.03, 1.28, 1.58, 1.94, and 2.27 were formulated. A protein mixture containing dehulled soybean meal, fish meal, wheat gluten flour, and chicken powder was used as the protein source. The crude lipid (CL) levels of all the treatments, which contained soybean lecithin, soybean oil, and fish oil, ranged from 122.4 g/kg to 188.0 g/kg. With increasing levels of the CHO:L ratio in the experimental diets, the proportion of corn starch was increased correspondingly to maintain the energy content in all the diets. All the powdered raw materials were ground to a fine powder, sieved through a 60 mesh screen, and then mixed thoroughly (M-256, South China University of Technology, Guangzhou, China). The distilled water and lipids were then added and stirred until well mixed. Pellet feed with a diameter of 2.5 mm was made (G-250, South China University of Technology,

Guangzhou, China) and air dried (air-conditioned room at 25 °C). The pellets were stored at −20 °C until they were used.

**Table 1.** Formulation and proximate composition of diets (g/kg, dry matter).

| Ingredients | Level of CHO:L Ratios in Diet | | | | | |
|---|---|---|---|---|---|---|
| | **0.82** | **1.03** | **1.28** | **1.58** | **1.94** | **2.27** |
| Fish meal | 380.0 | 380.0 | 380.0 | 380.0 | 380.0 | 380.0 |
| Dehulled soybean meal | 80.0 | 80.0 | 80.0 | 80.0 | 80.0 | 80.0 |
| Wheat gluten flour | 140.0 | 140.0 | 140.0 | 140.0 | 140.0 | 140.0 |
| Chicken powder | 100.0 | 100.0 | 100.0 | 100.0 | 100.0 | 100.0 |
| Fish oil | 15.0 | 15.0 | 15.0 | 15.0 | 15.0 | 15.0 |
| Soybean oil | 58.2 | 47.4 | 36.6 | 25.8 | 15.0 | 4.2 |
| Soybean lecithin | 20.0 | 20.0 | 20.0 | 20.0 | 20.0 | 20.0 |
| Corn starch | 105.0 | 130.0 | 155.0 | 180.0 | 205.0 | 230.0 |
| Calcium dihydrogen phosphate | 15.0 | 15.0 | 15.0 | 15.0 | 15.0 | 15.0 |
| Vitamin premix [1] | 3.0 | 3.0 | 3.0 | 3.0 | 3.0 | 3.0 |
| Mineral premix [1] | 7.0 | 7.0 | 7.0 | 7.0 | 7.0 | 7.0 |
| Vitamin C | 0.5 | 0.5 | 0.5 | 0.5 | 0.5 | 0.5 |
| Choline chloride | 3.0 | 3.0 | 3.0 | 3.0 | 3.0 | 3.0 |
| Ethoxyquin | 0.3 | 0.3 | 0.3 | 0.3 | 0.3 | 0.3 |
| Attractant | 1.0 | 1.0 | 1.0 | 1.0 | 1.0 | 1.0 |
| Carboxymethyl Cellulose | 1.0 | 1.0 | 1.0 | 1.0 | 1.0 | 1.0 |
| Cellulose | 71.0 | 56.8 | 42.6 | 28.4 | 14.2 | 0.0 |
| Total | 1000.0 | 1000.0 | 1000.0 | 1000.0 | 1000.0 | 1000.0 |
| Proximate composition [2] | | | | | | |
| Crude protein | 504.9 | 506.8 | 502.5 | 503.8 | 505.6 | 505.7 |
| Crude lipid | 188.0 | 172.6 | 159.4 | 144.8 | 130.7 | 122.4 |
| Nitrogen-free extract [3] | 153.4 | 178.4 | 203.4 | 222.7 | 253.4 | 278.4 |
| Gross energy (MJ/Kg) | 200.3 | 196.5 | 192.8 | 188.1 | 185.7 | 182.4 |

[1] Vitamin and mineral premix were provided by Qingdao Master Biotech Co., Ltd., Qingdao, China. [2] Proximate compositions were measured values. [3] Nitrogen-free extract was calculated by the difference.

### 2.2. Experimental Fish and Experimental Conditions

The experimental rearing was carried out for 8 weeks with juvenile hybrid groupers from a commercial fish hatchery (Hainan, China). Prior to the start of the feeding experiment, the juvenile hybrid groupers were conditioned to the culture environment (cement tank of size 2.4 m × 3 m ×2 m). The groupers were fed to apparent satiety twice a day with commercially available feed (Zhanjiang Shangshang Biological Feed Co., Ltd., Zhanjiang, China). During the 14-day acclimatization process, 70% of the water in the cement tank was replaced daily to ensure the safety of the culture environment. Groupers with an initial weight of 7.76 ± 0.08 g were randomly sorted into 24 fiberglass tanks (1000 L) of 30 fish each. The feeding experiments were conducted in a flow-through culture using indoor culture buckets with continuous oxygenation of the culture water. The photoperiod was the natural photoperiod. Feed was delivered twice daily at 8:00 and 16:00 to apparent satiety for 56 days. Seventy percent of the cultured water in each bucket was replaced each morning to ensure a stable environment for the cultured water. The feeding trial rearing conditions were maintained to keep the dissolved oxygen levels at 5 mg/L or higher, salinity at 27–29 g/L, ammonia nitrogen at a level below 0.03 mg/L, and water temperature at 28–30 °C.

### 2.3. Sample Collection and Analysis

Thirty fish from the initial groupers were randomly collected and stored at −20 °C before the start of the feeding experiment. At the end of the feeding trial, all the groupers were anesthetized with eugenol (1:10,000). The fish in each tank were weighed and counted

to determine the final body weight (FBW), protein efficiency ratio (PER), feed conversion ratio (FCR), specific growth rate (SGR), weight gain rate (WGR), and survival rate (SR). Three fish were taken at random from each tank to calculate the condition factor (CF), and the liver and viscera were taken and weighed separately to calculate the hepatopancreas somatic indices (HSI) and viscera somatic index (VSI). Three fish were chosen randomly from each dietary replicate and frozen at −20 °C to determine the lipid deposition rate (LDR) and protein deposition rate (PDR) and analyze the proximate composition. Liver and muscle tissue were collected randomly from 3 fish per tank and stored at −20 °C to analyze their proximate composition. Crude protein (CP), CL, moisture, ash, and crude fiber content were analyzed by AOAC [36]. The nitrogen-free extract (NFE) of the feed was calculated using a residual value. The glycogen contents were determined using liver and muscle glycogen assay kits (Nanjing Jiancheng Bioengineering Institute, Nanjing, China). The intestines and livers of 3 fish per tank were collected, placed in a paraformaldehyde solution, then stained (hematoxylin and eosin, and oil red), sectioned, and observed for histology. The embedding and staining of the intestinal and liver tissues were conducted by Wuhan Xavier Biotechnology Co. (Wuhan, China) to observe the lipid distribution in the liver and the muscle thickness (MT), villus width (VW), and villus length (VL) of the intestine.

After blood from 7 fish in each tank was collected by 1 mL sterile syringe into 1.5 mL Eppendorf tubes, the tubes were left to stand for 12 h at 4 °C, centrifuged (3500 rpm, 10 min), and the supernatant was stored at −80 °C for analysis of the serum biochemical index. The contents of serum total protein (TP), total cholesterol (CHOL), total triglyceride (TG), and glucose (GLU) were analyzed by an automatic chemistry analyzer Hitachi 7020 (Hitachi, Tokyo, Japan).

The intestine and liver were obtained from 7 fish immediately after the blood was collected, frozen in liquid nitrogen in a 2 mL frozen tube, and stored at −80 °C to conduct subsequent enzyme activity analysis. The activities of amylase, lipase, and trypsin in the liver and intestine were determined using kits (Nanjing Jiancheng Bioengineering Institute, Nanjing, China). The carbohydrate metabolism enzyme activities in liver, such as hexokinase (HK), 6-phosphofructokinase-1 (PFK-1), glycogen phosphorylase (GP), glycogen synthase (GS), phosphoenolpyruvate carboxy kinase (PEPCK), and pyruvate kinase (PK), and the lipid metabolism enzyme activities in liver, such as carnitine palmitoyl transferase (CPT), hepatic lipase (HL), lipoprotein lipase (LPL), hormone-sensitive triglyceride lipase (HSL), adipose triglyceride lipase (ATGL), and acetyl-CoA carboxylase (ACC) were analyzed with ELISA kits (Shanghai Zylian Biotechnology Co., Ltd., Shanghai, China) according to the manufacturer's instructions.

*2.4. Calculations and Statistical Analysis*

The formulas for SGR, WGR, FCR, PER, SR, CF, HSI, VSI, PDR, and LDR were referenced from our previous study [37]. In this study, the data were shown as mean ± standard error. Data were evaluated for normality of residuals (Shapiro-Wilk test) and homogeneity of variance (Bartlett test). Statistical verification of the data was subjected to a one-way analysis of variance ANOVA, and significant differences between the treatments were determined using Tukey's multiple comparison test (SPSS 20.0, IL, USA). The means were separated using linear and quadratic contrasts conducted with coefficients of unequal intervals of CHO:L ratios. $p < 0.05$ was considered significant. In addition, a second-order polynomial regression model was fitted using GraphPad Prism (8.0.1) based on the SGR to determine the optimal CHO:L ratio in the diet.

## 3. Results

*3.1. Growth Performance and Body Morphologic*

The SR, FBW, SGR, WGR, PER, FCR, PDR, and CF of the groupers were not significantly affected by the feed ($p > 0.05$). The LDR, HSI, and VSI of the groupers showed a significant decrease as the dietary CHO:L ratios increased ($p < 0.05$). The LDR, HSI,

and VSI were significantly higher in the 0.82 group than those in the 2.27 group ($p < 0.05$; Table 2). In this experiment, the SGR was used as an evaluation indicator, and the relationship between the SGR of the groupers and dietary CHO:L ratios was fitted by the second-order polynomial regression model. The most appropriate dietary CHO:L ratio was 1.72 (Figure 1).

**Table 2.** Growth performance and body morphologic indexes of hybrid groupers.

| Index | Dietary CHO:L Ratios | | | | | | Pooled SEM | *p*-Values | | |
| --- | --- | --- | --- | --- | --- | --- | --- | --- | --- | --- |
| | 0.82 | 1.03 | 1.28 | 1.58 | 1.94 | 2.27 | | Treatment | Linear | Quadratic |
| IBW (g) | 7.74 | 7.80 | 7.69 | 7.74 | 7.71 | 7.92 | 0.12 | 0.772 | 0.465 | 0.464 |
| FBW (g) | 56.57 | 57.44 | 57.32 | 63.56 | 57.33 | 58.08 | 2.35 | 0.364 | 0.626 | 0.425 |
| WGR (%) | 611.72 | 646.02 | 664.07 | 725.45 | 687.86 | 654.22 | 33.21 | 0.317 | 0.363 | 0.061 |
| SGR (%/d) | 3.50 | 3.59 | 3.63 | 3.77 | 3.69 | 3.60 | 0.07 | 0.280 | 0.342 | 0.046 |
| FCR | 0.96 | 1.01 | 1.00 | 0.99 | 1.02 | 1.07 | 0.02 | 0.055 | 0.012 | 0.521 |
| PER | 0.97 | 0.98 | 0.99 | 1.11 | 0.98 | 1.00 | 0.05 | 0.362 | 0.665 | 0.400 |
| SR (%) | 95.84 | 95.00 | 94.17 | 97.34 | 97.34 | 95.84 | 0.40 | 0.162 | 0.184 | 0.071 |
| PDR (%) | 39.45 | 38.30 | 39.51 | 38.31 | 38.93 | 36.95 | 0.41 | 0.547 | 0.200 | 0.507 |
| LDR (%) | 43.39 [a] | 40.50 [b] | 39.98 [b] | 40.92 [b] | 39.00 [b] | 39.29 [b] | 0.38 | <0.001 | <0.001 | 0.026 |
| HSI (%) | 1.79 [a] | 1.69 [ab] | 1.64 [ab] | 1.48 [abc] | 1.13 [bc] | 1.00 [c] | 0.08 | 0.006 | <0.001 | 0.280 |
| VSI (%) | 9.51 [a] | 8.69 [ab] | 8.85 [ab] | 7.84 [ab] | 8.07 [ab] | 7.53 [b] | 0.21 | 0.033 | 0.002 | 0.639 |
| CF (%) | 2.67 | 2.80 | 2.86 | 3.00 | 2.84 | 2.82 | 0.04 | 0.221 | 0.165 | 0.055 |

Values are means from each group of fish (*n* = 4). In the same row, values with no letter or the same letter superscripts mean no significant difference ($p > 0.05$), while those with different lowercase letters mean a significant difference ($p < 0.05$). IBW, initial body weight; FBW, final body weight; WGR, weight gain rate; SGR, specific growth rate; FCR, feed conversion rate; PER, protein efficiency ratio; SR, survival rate; PDR, protein deposition rate; LDR, lipid deposition rate; HSI, hepatopancreas somatic indices; VSI, viscera somatic index; CF, condition factor.

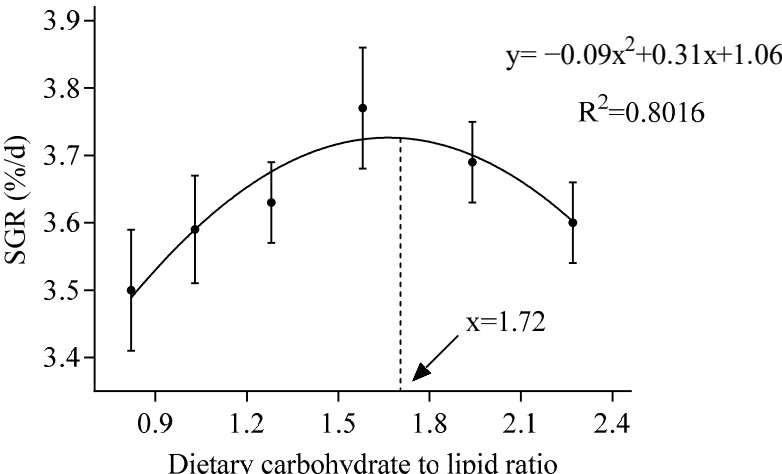

**Figure 1.** Relationship between dietary CHO:L ratios and SGR of hybrid grouper was fitted by the second-order polynomial regression model. Mean values ± SEM are presented for each group (*n* = 4).

### 3.2. Whole Body Composition

The CP and ash contents of the whole body were not significantly affected by the dietary CHO:L ratio ($p > 0.05$). With the dietary CHO:L ratio increased, the moisture content showed a significantly increased trend and reached a maximum in the 2.27 group ($p < 0.05$). The CL content of the whole fish decreased significantly with the increasing dietary CHO:L ratio. Significantly lower CL levels were found in the 1.58, 1.94, and 2.27 groups than in the 0.82 group ($p < 0.05$, Table 3).



**Table 3.** Whole body composition of hybrid groupers (g/kg, dry matter).

| Index | Dietary CHO:L Ratios | | | | | | Pooled SEM | p-Values | | |
|---|---|---|---|---|---|---|---|---|---|---|
| | 0.82 | 1.03 | 1.28 | 1.58 | 1.94 | 2.27 | | Treatment | Linear | Quadratic |
| Moisture (wet) | 691.9 b | 697.7 ab | 700.3 ab | 700.6 ab | 701.6 ab | 707.5 a | 0.15 | 0.038 | 0.002 | 0.827 |
| Crude protein | 565.8 | 570.2 | 574.9 | 571.2 | 572.9 | 581.7 | 0.26 | 0.315 | 0.171 | 0.876 |
| Crude lipid | 346.4 a | 315.4 ab | 312.1 ab | 304.0 b | 292.3 b | 297.2 b | 0.50 | 0.666 | <0.001 | 0.055 |
| Ash | 139.7 | 143.6 | 158.7 | 155.0 | 152.5 | 153.1 | 0.26 | 0.273 | 0.092 | 0.134 |

Values are means from each group of fish ($n = 4$). In the same row, values with no letter or the same letter superscripts mean no significant difference ($p > 0.05$), while with different lowercase letters mean a significant difference ($p < 0.05$).

### 3.3. Ether Extract and Glycogen Contents in Liver and Muscle

The CL content of the liver was significantly lower in the 2.27 group than in the 0.82 group ($p < 0.05$). With the increase in the CHO:L ratio, the liver glycogen content and muscle CL content showed a change of first increased and then decreased, both of which were significantly lowest in the 2.27 group ($p < 0.05$). The glycogen content in the muscle was not significantly changed by the diet ($p > 0.05$, Table 4). The liver oil red staining slices showed a gradual decrease in lipid content as the CHO:L ratio increased (Figure 2).

**Table 4.** The ether extract and glycogen in liver and muscle of hybrid groupers.

| Index (g/kg) | Dietary CHO:L Ratios | | | | | | Pooled SEM | p-Values | | |
|---|---|---|---|---|---|---|---|---|---|---|
| | 0.82 | 1.03 | 1.28 | 1.58 | 1.94 | 2.27 | | Treatment | Linear | Quadratic |
| Liver | | | | | | | | | | |
| Crude lipid | 586.4 a | 569.6 ab | 576.0 ab | 536.1 ab | 514.2 ab | 485.4 b | 2.53 | 0.015 | 0.002 | 0.009 |
| Glycogen | 273.3 ab | 377.7 a | 308.5 ab | 301.0 ab | 283.5 ab | 230.1 b | 1.38 | 0.027 | 0.033 | 0.026 |
| Muscle | | | | | | | | | | |
| Crude lipid | 390.1 c | 410.0 bc | 461.8 a | 458.3 a | 430.0 b | 387.0 c | 0.68 | 0.001 | 0.869 | <0.001 |
| Glycogen | 13.2 | 13.0 | 13.9 | 13.2 | 13.3 | 12.0 | 0.03 | 0.738 | 0.428 | 0.272 |

Values are means from each group of fish ($n = 4$). In the same row, values with no letter or the same letter superscripts mean no significant difference ($p > 0.05$), while with different lowercase letters mean a significant difference ($p < 0.05$).

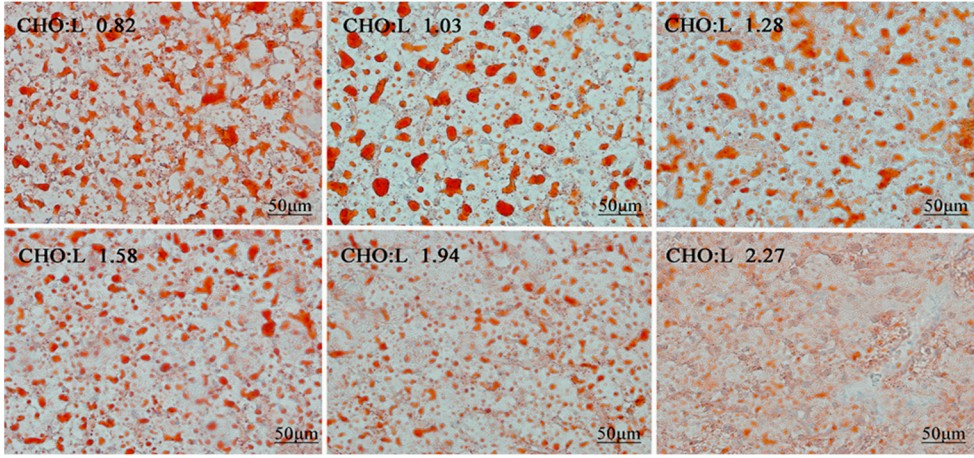

**Figure 2.** The liver oil red staining sections of hybrid groupers fed with diets. Light microscopy: 40×. Scale bar: 50 μm.

### 3.4. Liver and Intestinal Digestive Enzyme Activities

In the 1.58 group, the activities of trypsin and lipase in the intestine were significantly greater than the others ($p < 0.05$). As the dietary CHO:L ratio increased, intestinal amylase activity decreased significantly, and the amylase activity showed a minimum value in the

2.27 group ($p < 0.05$). In the liver, the trypsin activity in the 1.58 group was significantly more than in all other groups ($p < 0.05$), and the lipase and amylase activities in the 1.28 group were significantly higher than those in the 1.58, 1.94, and 2.27 groups ($p < 0.05$, Table 5).

**Table 5.** Intestinal and liver digestive enzyme activities of hybrid groupers.

| Index (U/mg Protein) | Dietary CHO:L Ratios | | | | | | Pooled SEM | *p*-Values | | |
|---|---|---|---|---|---|---|---|---|---|---|
| | 0.82 | 1.03 | 1.28 | 1.58 | 1.94 | 2.27 | | Treatment | Linear | Quadratic |
| Intestine | | | | | | | | | | |
| Trypsin | 2467.20 bc | 1443.87 d | 2910.72 b | 3718.29 a | 2405.41 bc | 2248.73 c | 169.45 | 0.001 | 0.712 | 0.219 |
| Lipase | 5.55 d | 9.26 c | 12.90 b | 15.47 a | 8.57 c | 4.52 e | 0.27 | <0.001 | 0.544 | <0.001 |
| Amylase | 0.25 a | 0.21 b | 0.18 c | 0.17 c | 0.15 d | 0.12 e | 0.006 | <0.001 | <0.001 | <0.001 |
| Liver | | | | | | | | | | |
| Trypsin | 1593.69 c | 1677.94 c | 1596.91 c | 2733.20 a | 2175.07 b | 830.77 d | 72.96 | <0.001 | 0.548 | 0.008 |
| Lipase | 3.59 a | 2.00 bc | 3.52 a | 2.54 b | 1.57 cd | 1.13 d | 0.21 | <0.001 | 0.005 | 0.019 |
| Amylase | 0.10 b | 0.14 a | 0.15 a | 0.10 b | 0.11 b | 0.11 b | 0.01 | <0.001 | 0.569 | 0.141 |

Values are means from each group of fish ($n = 4$). In the same row, values with no letter or the same letter superscripts mean no significant difference ($p > 0.05$), while with different lowercases mean a significant difference ($p < 0.05$).

### 3.5. Intestinal Morphology

From Table 6 and Figure 3, the VL in the 1.58 group was significantly higher than that in other groups ($p < 0.05$). The VW in the 2.27 group was not significantly different from the 1.94 group but was significantly below that in all other groups ($p < 0.05$). In the 1.94 group, MT was not significantly different from the 1.58 group but was significantly greater than in the other groups ($p < 0.05$).

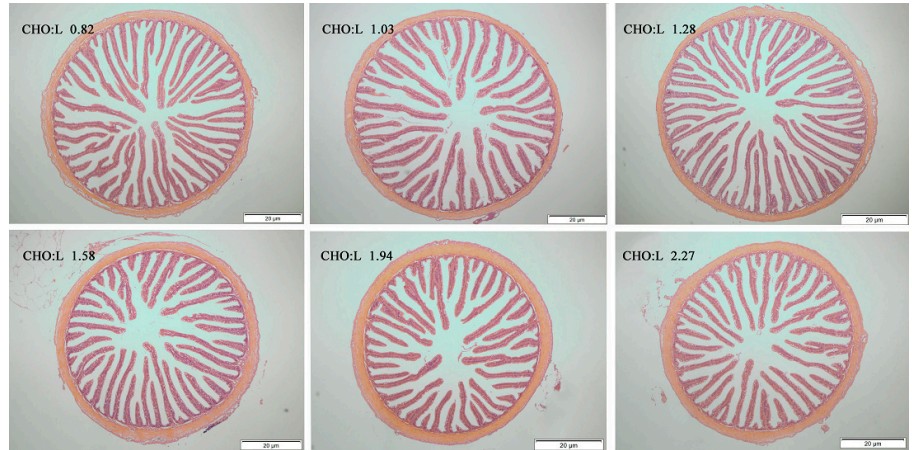

**Figure 3.** The intestine morphology hematoxylin-eosin staining of hybrid groupers. Light microscopy: 4×. Scale bar: 20 μm.

**Table 6.** Intestinal morphology of hybrid groupers.

| Index (μm) | Dietary CHO:L Ratios | | | | | | Pooled SEM | *p*-Values | | |
|---|---|---|---|---|---|---|---|---|---|---|
| | 0.82 | 1.03 | 1.28 | 1.58 | 1.94 | 2.27 | | Treatment | Linear | Quadratic |
| VL | 264.14 bc | 284.20 b | 253.60 bc | 309.68 a | 284.33 b | 238.83 c | 13.83 | <0.001 | 0.598 | <0.001 |
| VW | 36.87 a | 37.67 a | 34.21 a | 32.34 a | 31.67 ab | 25.64 b | 1.07 | 0.001 | <0.001 | 0.091 |
| MT | 37.58 c | 38.57 c | 35.89 c | 49.98 ab | 55.29 a | 49.52 b | 1.84 | <0.001 | <0.001 | 0.869 |

Values are means from each group of fish ($n = 4$). In the same row, values with no letter or the same letter superscripts mean no significant difference ($p > 0.05$), while with different lowercases mean a significant difference ($p < 0.05$). VL, villus length; VW, villus width; MT, muscle thickness.

*3.6. Serum Biochemical Index*

The serum TP contents were not significantly affected by diets ($p > 0.05$). The GLU content in the 1.28 and 2.27 groups was significantly greater than that in the 0.82 group ($p < 0.05$). In the 2.27 group, serum TG content was lower than that in the other groups ($p < 0.05$). In the 0.82 and 1.28 groups, CHOL content was significantly higher than that in the 1.58, 1.94, and 2.27 groups ($p < 0.05$, Table 7).

**Table 7.** Serum biochemical index of hybrid groupers.

| Index (mmol/L) | Dietary CHO:L Ratios | | | | | | Pooled SEM | *p*-Values | | |
|---|---|---|---|---|---|---|---|---|---|---|
| | 0.82 | 1.03 | 1.28 | 1.58 | 1.94 | 2.27 | | Treatment | Linear | Quadratic |
| TP | 49.77 | 51.03 | 50.57 | 50.76 | 51.34 | 49.62 | 0.87 | 0.684 | 0.898 | 0.344 |
| GLU | 4.78 [b] | 6.87 [ab] | 8.53 [a] | 8.01 [ab] | 8.11 [ab] | 8.64 [a] | 1.02 | 0.023 | 0.031 | 0.024 |
| TG | 1.03 [a] | 0.84 [a] | 0.80 [a] | 0.91 [a] | 0.81 [a] | 0.53 [b] | 0.08 | 0.048 | 0.011 | 0.033 |
| CHOL | 4.3 [a] | 3.73 [ab] | 4.01 [a] | 3.01 [bc] | 2.65 [cd] | 2.10 [d] | 0.25 | 0.001 | <0.001 | <0.001 |

Values are means from each group of fish ($n = 4$). In the same row, values with no letter or the same letter superscripts mean no significant difference ($p > 0.05$), while with different lower cases mean a significant difference ($p < 0.05$). TP, total protein; GLU, glucose; TG, total triglyceride; CHOL, total cholesterol.

*3.7. Hepatic Enzyme Activities of Carbohydrate Metabolism*

As the dietary CHO:L ratio increased, the activities of HK, PFK-1, PK, and GS showed a trend of first increasing and then decreasing. The activity of HK was significantly higher in the 1.58 and 1.94 groups than in other groups ($p < 0.05$). A significant maximum of HK activity was shown in the 1.94 group ($p < 0.05$). In the 1.28 and 1.58 groups, PK activity was significantly higher than that in other groups ($p < 0.05$). The GS activity in the 0.82 and 2.27 groups was significantly lower than that in other groups ($p < 0.05$). As the dietary CHO:L ratio increased, the activity of GP tended to decrease and then increase, with a minimum value in the 1.58 group ($p < 0.05$), the activity of PEPCK showed a decreasing pattern, with the lowest PEPCK activity significantly in the 2.27 group ($p < 0.05$, Table 8).

**Table 8.** Enzyme activities of carbohydrate metabolism in liver of hybrid groupers.

| Index | Dietary CHO:L Ratios | | | | | | Pooled SEM | *p*-Values | | |
|---|---|---|---|---|---|---|---|---|---|---|
| | 0.82 | 1.03 | 1.28 | 1.58 | 1.94 | 2.27 | | Treatment | Linear | Quadratic |
| HK (U/g protein) | 0.0048 [c] | 0.0057 [b] | 0.0060 [b] | 0.0070 [a] | 0.0073 [a] | 0.0059 [b] | 0.0002 | <0.001 | 0.023 | <0.001 |
| PFK-1 (U/g protein) | 131.95 [d] | 147.21 [c] | 160.47 [b] | 165.73 [b] | 177.85 [a] | 128.34 [d] | 1.73 | <0.001 | 0.193 | <0.001 |
| PK (mU/g protein) | 2.26 [d] | 2.51 [b] | 3.16 [a] | 3.32 [a] | 2.76 [b] | 2.38 [cd] | 0.06 | <0.001 | 0.875 | <0.001 |
| GP (U/g protein) | 0.93 [a] | 0.84 [b] | 0.76 [c] | 0.66 [d] | 0.86 [b] | 0.94 [a] | 0.02 | <0.001 | 0.681 | <0.001 |
| GS (U/g protein) | 2.53 [d] | 3.62 [a] | 3.37 [b] | 3.77 [a] | 3.63 [a] | 2.87 [c] | 0.06 | <0.001 | 0.654 | <0.001 |
| PEPCK (U/g protein) | 0.74 [a] | 0.71 [a] | 0.56 [b] | 0.52 [b] | 0.55 [b] | 0.45 [c] | 0.01 | <0.001 | 0.015 | 0.055 |

Values are means from each group of fish ($n = 4$). In the same row, values with no letter or the same letter superscripts mean no significant difference ($p > 0.05$), while with different lowercases mean a significant difference ($p < 0.05$). HK, hexokinase; PFK-1, 6-phosphofructokinase-1; PK, pyruvate kinase; GP, glycogen phosphorylase; GS, glycogen synthase; PEPCK, phosphoenolpyruvate carboxy kinase.

*3.8. Hepatic Enzyme Activities of Lipid Metabolism*

As the dietary CHO:L ratio increased, CPT activity increased significantly and gradually, with a maximum in the 1.58 and 2.27 groups ($p < 0.05$), ATGL, HSL, and ACC activity showed an increase followed by a decrease, in the 1.94 group, the activity of TGL, HSL, and ACC was significantly greatly than that in the 0.82, 1.03 and 2.27 groups ($p < 0.05$). HL and LPL activity in the 0.82 and 2.27 groups was significantly higher than that in the 1.58 group ($p < 0.05$, Table 9).

**Table 9.** Enzyme activities of lipid metabolism in liver of hybrid groupers.

| Index (U/g protein) | Dietary CHO:L Ratios | | | | | | Pooled SEM | *p*-Values | | |
|---|---|---|---|---|---|---|---|---|---|---|
| | 0.82 | 1.03 | 1.28 | 1.58 | 1.94 | 2.27 | | Treatment | Linear | Quadratic |
| CPT | 1.98 [d] | 2.73 [c] | 3.64 [b] | 3.96 [a] | 4.05 [a] | 3.91 [a] | 0.05 | <0.001 | <0.001 | <0.001 |
| ATGL | 3.57 [c] | 3.80 [b] | 4.01 [a] | 4.12 [a] | 3.99 [a] | 3.40 [c] | 0.06 | <0.001 | 0.715 | <0.001 |
| HSL | 5.75 [d] | 6.96 [c] | 8.34 [b] | 11.43 [a] | 11.46 [a] | 8.67 [b] | 0.17 | <0.001 | 0.005 | <0.001 |
| LPL | 23.54 [a] | 20.71 [ab] | 15.65 [ab] | 13.60 [b] | 17.51 [ab] | 23.18 [a] | 2.24 | 0.007 | 0.888 | 0.005 |
| HL | 31.40 [a] | 27.91 [ab] | 29.18 [ab] | 16.20 [c] | 25.79 [ab] | 24.03 [b] | 1.96 | 0.006 | 0.140 | 0.056 |
| ACC | 240.01 [e] | 275.30 [d] | 302.05 [c] | 341.42 [b] | 371.14 [a] | 248.19 [e] | 3.54 | <0.001 | 0.273 | <0.001 |

Values are means from each group of fish (*n* = 4). In the same row, values with no letter or the same letter superscripts mean no significant difference (*p* > 0.05), while with different lowercases mean significant difference (*p* < 0.05). CPT, carnitine palmitoyl transferase; ATGL, adipose triglyceride lipase; HSL, hormone-sensitive triglyceride lipase; LPL, lipoprotein lipase; HL, hepatic lipase; ACC, acetyl-CoA carboxylase.

## 4. Discussion

The experimental results of our study showed that dietary CHO: L ratios ranging from 0.82 to 2.27 did not significantly affect the SR and growth performance of grouper. When the CHO:L ratio was increased from 0.82 to 1.58, the WGR, SGR, and PER of the hybrid grouper showed an upward trend; after that, they showed a downward change. Similar findings have been reported on fish with various feeding habits, such as herbivorous fish, e.g., grass carp (*Ctenopharyngodon idella*) [38], omnivorous fish, e.g., tilapia (*Oreochromis niloticus*) [39], Brazilian sardines (*Sardinella brasiliensis*) [40], and yellowfin seabream (*Sparus latus*) [41], and carnivorous fish, e.g., golden pompano (*Trachinotus ovatus*) [24], orange-spotted grouper (*E. coioides*) [26], early giant grouper (*E. Lanceolatus*) [42], and walking catfish (*Clarias batrachus*) [43]. This may be due to the synergistic effects of the carbohydrates and lipids being fully exerted, thereby improving the growth performance of the fish at an appropriate feed CHO:L ratio [44]. In this research, the growth performance showed a numerically increasing trend with an increasing CHO:L ratio based on isonitrogenous and isoenergetic diets. This implies that hybrid grouper may be better able to utilize carbohydrates despite its carnivorous habits. Excessively high lipid or carbohydrate content in the feed reduces the palatability of the feed, thereby reducing the feeding of the fish, and limiting their growth performance [45]. A similar situation was found in this experiment. The growth performance of the groupers showed a numerically decreasing pattern when feeds with too-high CHO:L ratios were consumed. In addition, the numerical reduction in the growth performance of the groupers may also be due to a deficiency and imbalance of essential fatty acids as a result of reduced soybean oil content in the diet.

In this study, the relationship between the SGR of the groupers and dietary CHO:L ratios was fitted through the second-order polynomial regression model. The results of our experiment revealed that the optimum dietary CHO:L ratio for the hybrid grouper (initial body weight 7.76 ± 0.08 g) was 1.72. However, another study showed that the optimal CHO:L ratio (initial weight was 21.48 ± 0.24 g) for hybrid grouper was 0.77, based on SGR (CP was 520.8 g/kg, CL was 103.7 g/kg, and NFE was 80.2 g/kg) [46]. This may be related to the differences in the initial body weight of the experimental fish, culture environment, and feed formulation [47]. Studies have found that the optimal CHO:L ratio for carnivorous fish, such as early giant grouper, was 1.30 [42], large yellow croaker was 1.34 [8], Chinese longsnout catfish (*Leiocassis longirostris* Günther) was 1.98, [48], and tropical gar was 2.10 [49]. These results were lower than the optimal CHO:L ratio of omnivorous fish, such as Brazilian sardines, which was 3.41 [40], and herbivorous fish, such as blunt snout bream (*Megalobrama amblycephala*), which was 3.58 [50], and herbivorous grass carp, which was 4.70 [38]. Carnivorous fish are generally characterized by a weak tolerance to carbohydrates, which is also the reason why carnivorous fish feeds have lower optimal CHO:L ratios [51].

As an important indicator for evaluating the nutritional efficiency of feeds, FCR is important for assessing the utilization of feeds in grouper. For juvenile fish, the change of FCR should be more sensitive. However, in this experiment, the FCR of the groupers did

not change significantly with the dietary CHO: L ratio. Similar situations were observed in white seabass (*Atractoscion nobilis*, Ayres 1860; initial body weight was 9.5 ± 0.1 g) [23], tropical gar (*Atractosteus tropicus*; initial body weight was 0.50 ± 0.01 g) [49], large yellow croaker (*Larmichthys crocea*; initial body weight was 7.06 ± 0.48 g) [8], and silvery-black porgy (*Sparidentex hasta*; initial body weight was 14.60 ± 0.10 g).

The dietary lipid level is associated with the body's lipid deposition [52]. Dietary lipids in excess may cause excessive deposition of lipids in muscle, liver, and mesentery tissues [19]. This phenomenon was supported by the results of LDR and the CL contents of the whole body, liver, and muscle in our experiment. With the CHO:L ratios increased, the CL contents of the whole body and liver showed a decreasing trend, which was compatible with the results of Brazilian sardines [40], large yellow croaker [8], rockfish (*Sebastes schlegeli*) [53], and Chinese longsnout catfish [48]. The CL contents of the muscle showed a change of an increase and then a decrease, which was consistent with the research results of blunt snout bream [50] and rockfish [53]. The HSI and VSI decreased significantly with the increasing CHO:L ratio, which was consistent with the results of silvery-black porgy [28] and herbivorous grass carp [38]. The decrease in intrahepatic CL content was an important factor in the decrease in HSI.

In our study, appropriate levels of dietary CHO:L ratios significantly increased the intestinal VL and MT. This was directly related to the improvement in growth performance and the intestinal digestive enzyme activities in the hybrid grouper. With the increasing CHO:L ratios, the change trend in the trypsin activities of the intestine and liver were consistent with the change trends in the WGR, SGR, and PER, and both showed the maximum value in the 1.58 group. This may be because this dietary CHO:L ratio could keep the activities of the trypsin at a high level, enhance the utilization of protein, and improve the growth performance of the hybrid grouper. As the CHO:L ratio increases, the intestine lipase activity reaches the maximum in the 1.58 group. The appropriate lipid level of the diet can promote the secretion of lipase in the intestine, increase the contact area of the lipase and substrate, and increase the digestive capacity of the lipids [54,55]. However, with the increase in the CHO:L ratios, the activities of liver lipase gradually decreased, which was consistent with the research results on golden pompano [24] and large yellow croaker [8]. The activity of liver amylase showed a pattern of increasing first and then decreasing. Optimal carbohydrate levels can improve the hepatic amylase activity of golden pompano, *Pseudoplatystoma corruscans*, and *sea bass* (*Dicentrarchus labrax*) [24,56,57].

The liver, as a consumer and producer of glucose, plays a central role in controlling glucose homeostasis [58]. HK, PFK-1, and PK are the three key rate-limiting enzymes in the glycolysis pathway; to a certain extent, their activities can reflect the carbohydrate metabolism of fish [59,60]. In this experiment, when the dietary CHO:L ratios increased from 0.82 to 1.58 or 1.94, the activities of the HK, PFK-1, and PK were significantly increased, and the glycolysis was enhanced, indicating that when the dietary CHO:L ratios were appropriate, they facilitate the utilization of carbohydrates by the grouper and promote its rapid growth. Enhanced glycolysis resulted in a massive breakdown of carbohydrates, providing more energy to meet the growth and metabolic needs of the grouper. Adequate energy supply can reduce the occurrence of the grouper utilizing its own protein for energy supply. When the dietary CHO:L ratios further increased, the activities of HK, PFK-1, and PK were significantly reduced. From 222.7 g/kg to 253.4 g/kg of dietary NFE, the tolerance to carbohydrates of the hybrid grouper can be limited. GS is activated by phosphorylation reactions and used to catalyze the synthesis of glycogen [61]. In this experiment, with the increase in the CHO:L ratios, the GS activity showed a change of rising and then falling. At the appropriate CHO:L ratio, the GS activity was promoted, and the ability of the hybrid grouper to utilize carbohydrates was improved. Glucose synthesizes glycogen through the GS catalysis and then stores it in the liver and muscle. In our study, the change trend in the glycogen contents in the liver was consistent with the change trend in the GS activity. When the CHO:L ratio was optimal, the enzyme activities of HK, PFK-1, PK, and GS were increased significantly, and more glucose could be converted into a series of metabolites

and glycogen. However, this function was far from adequate compared to the carbohydrate intake, which leads to the serum glucose contents varying with the feed's CHO:L ratio rise [49]. Similar situations were found in large yellow croaker [8,25] and orange-spotted grouper [62].

GP is activated by dephosphorylation reactions, used to catalyze the decomposition of glycogen [63]. The pattern of GP activities was opposite to those of the GS activities in the hybrid grouper liver. PEPCK, the key enzyme that controls gluconeogenesis [64], can synthesize glucose and glycogen using non-carbohydrates such as lactic acid, glycerin, and keto acid [65]. When the CHO:L ratio in the feed was high, the hepatic PEPCK activity of the hybrid grouper was significantly reduced. The increase in GP activity and the decrease in PEPCK activity were important reasons for the significant reduction in the liver glycogen contents at high CHO:L ratios, and a similar result was obtained in large yellow croaker [8].

CPT and HSL are the rate-limiting enzymes for fatty acid beta-oxidation [66] and lipolysis, respectively, and can be used as indicators for monitoring the lipid content of fish [67]. ATGL is the initial step in catalyzing the hydrolysis of triglycerides [68]. ACC is the rate-limiting enzyme for fatty acid synthesis and plays an important role in fatty acid synthesis metabolism [69]. When the feed's CHO:L ratio was optimal, the activities of enzymes involved in fatty acid oxidation (CPT), decomposition (HSL and ATGL), and fatty acid synthesis (ACC) were significantly increased, which improves the lipid metabolism and growth of the hybrid grouper, which was closely related to the significant reduction of serum CHOL contents. Lipids are broken down to form fatty acids and glycerol which release large amounts of energy after decomposition, metabolism and oxidation. These energies can be effectively utilized by groupers, thereby reducing the occurrence of using their own proteins for energy consumption. With the increase in dietary CHO:L ratios, the contents of serum TG and CHOL were significantly reduced, which may also be due to the reduction of lipid content in the diet and the reduction of free fatty acids absorbed and transformed by the fish, which reduces the CHOL and TG synthesized and transported by the liver, resulting in a significant reduction of CHOL and TG contents in serum [70]. Similar results were found on herbivorous grass carp [38], tilapia [39], yellow catfish (*Pelteobagrus fulvidraco*) [44], and yellowfin seabream [41].

LPL and HL, with a high degree of structural similarity, can regulate lipolysis and lipid absorption [71]. LPL is responsible for hydrolyzing triglycerides and provides free fatty acids for oxidation or storage in tissues [72]. HL is mainly synthesized in the liver and involved in the metabolism of chylomicron emulsion and high-density lipoproteins [73]. With proper feed CHO:L ratios, the less crude lipid in the feed, the less lipid content will be ingested. However, the activities of LPL and HL in the liver were reduced, which may be an important factor showing that the serum TG contents were not significantly affected. In addition, the LPL is a key enzyme that promotes lipid deposition [74]. With the increase in feed CHO:L ratios, the activity of LPL showed a trend of decreasing first and then increasing. This was consistent with the downward change in the LDR and CL contents in the whole body and liver. Although the LPL activity was increased at high CHO:L ratios, due to the lower lipid levels in the feed, more lipids were used in the growth of the fish, so that the lipid deposition was not increased [75].

## 5. Conclusions

Under the conditions of this experiment, based on the SGR of hybrid grouper, after fitting by the second-order polynomial regression model, the best dietary CHO:L ratio was 1.72. Appropriate dietary CHO:L ratios can enhance growth performance, reduce lipid deposition in the fish body, and improve intestinal morphology and the enzyme activities of digestion and carbohydrate and lipid metabolism.

**Author Contributions:** X.Y.: conceptualization, investigation, formal analysis, and writing—original draft. X.G.: conceptualization, investigation, and formal analysis. X.D.: methodology and resources. Q.Y.: methodology and resources. H.L.: methodology and resources. S.Z.: methodology and resources. B.T.: methodology, resources, and funding acquisition. S.C.: conceptualization, writing—review and editing, supervision, project administration, and funding acquisition. All authors have read and agreed to the published version of the manuscript.

**Funding:** This study was supported by the China Agriculture Research System of MOF and MARA (CARS-47), the Guangdong Provincial Higher Education Key Field Special Project (2020ZDZX1034), and the Zhanjiang Science and Technology Plan Project (2020A02001).

**Institutional Review Board Statement:** All the animal experiments were approved by the Animal Ethics Committee of Guangdong Ocean University (Zhanjiang, China; GDOU-01/2019) and were conducted in accordance with the recommendations in the Guide for the Care and Use of Laboratory Animals developed by the National Institutes of Health.

**Data Availability Statement:** The data that support the findings of this study are available on request from the corresponding author.

**Conflicts of Interest:** The authors declare no competing financial interest.

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
