# Peer review of "How Do Different Dietary Carbohydrate/Lipid Ratios Influence Intestinal Morphology and Glycolipid Metabolism Capacity in Hybrid Grouper (Epinephelus fuscoguttatus ♀ × E. lanceolatus ♂)"

_fishes, doi:10.3390/fishes8090467_

Round 1

Reviewer 1 Report (New Reviewer)

The manuscript presented to me for review is interesting and important from the aquaculture development point of view. In general, the work, in my opinion, does not require too many corrections and specific comments to the text can be found below. 

1. Lines 12 to 16 onwards in the text of the manuscript. The need to explain the sense of marking fragments in yellow.

2. Lines 33 and 34. Suggestion of combining keywords: "carbohydrate and lipid metabolism".

3. Line 173. Information referring to Figure 1 is given in the text and table 2 is given directly below. The need to change the formatting.

4. Lines 193, 218 and further below the tables. For example, below Table 3 there is information on the number of fish taken to calculate the average. There is doubt whether n=4 is sufficient to draw reliable conclusions. Please explain.

5. The need to check the consistency of the reference cited in the text with its list at the end of the manuscript.

Author Response

Reply to the Review Report

Reviewer 1

Comments and Suggestions for Authors

The manuscript presented to me for review is interesting and important from the aquaculture development point of view. In general, the work, in my opinion, does not require too many corrections and specific comments to the text can be found below.

Reply: Thank you very much for recognizing our work. We have carefully revised the manuscript as you suggested.

1. Lines 12 to 16 onwards in the text of the manuscript. The need to explain the sense of marking fragments in yellow.

Reply: Thank you very much for raising the query. In the previous manuscript, we started the introduction of the experiment straight away, without giving the background or hypothesis. We have revised it to address this issue. In addition, we have rewritten the section. That's why they are marked in yellow.

2. Lines 33 and 34. Suggestion of combining keywords: "carbohydrate and lipid metabolism".

Reply: Thank you very much for your suggestion. You suggestion makes our keywords more refined and precise. We have revised lines 32 and 33 of the manuscript as you suggested.

3. Line 173. Information referring to Figure 1 is given in the text and table 2 is given directly below. The need to change the formatting.

Reply: Thank you very much for your suggestions. Your suggestions have made our manuscript more accessible. We have adjusted the formatting of Figure 1 and Table 2 as you suggested.

4. Lines 193, 218 and further below the tables. For example, below Table 3 there is information on the number of fish taken to calculate the average. There is doubt whether n=4 is sufficient to draw reliable conclusions. Please explain.

Reply: Thank you very much for your question. In this experiment, we had a total of six treatment groups. In lines 110-111 of the manuscript, we mention “Randomly sorted groupers with an initial weight of 7.76 ± 0.08 g into 24 fiberglass tanks (1000 L) of 30 fish each”. From this, we know that each treatment group has 4 parallel replications. The “n” in the manuscript refers to the value of parallel repetitions, not the number of fish.

5. The need to check the consistency of the reference cited in the text with its list at the end of the manuscript.

Reply: Thank you very much for your suggestions. Your suggestions have made our manuscript more rigorous. We examined the references throughout the manuscript for consistency on a one-to-one basis.

Reviewer 2 Report (New Reviewer)

This ms examines the effects of dietary carbohydrate-to-lipid ratio (CHO:L) on growth performance, body composition, intestinal morphology, and glycolipid metabolism in juvenile hybrid grouper fish. The topic is relevant given the lack of specific nutritional guidelines for this emerging aquaculture species. The study design using 6 isonitrogenous, isoenergetic diets with graded CHO:L ratios is appropriate. The data appears to be thoroughly collected and analyzed. However, some issues need to be addressed:

Abstract:

- Line 20-21: Please clarify if you meant "specific growth rate" here.

- Line 29-30: The statement about the quadratic regression model is redundant since it is mentioned earlier. Consider removing.

M&M:

- Please add details on the feeding strategy during acclimatization.

- On lines 123 and 179, "protein efficiency rate" is used to describe the protein efficiency ratio (PER) calculation. PER is the standard term for this metric in the aquaculture nutrition literature. I suggest revising the text on lines 123 and 179 to use the terminology "protein efficiency ratio (PER)" rather than protein efficiency rate. This edit will align the terminology with the accepted standard in the field.

- Expand section 2.4 to provide more information on the regression models and variables used in the analysis.

Discussion:

- Consider explaining the potential mechanisms and biological significance of key findings. For example, the increase in glycolytic enzymes at intermediate CHO:L ratios likely reflects greater carbohydrate catabolism and energy production. This context would add helpful physiological insights.

- Summarize the main points to improve flow and reduce redundancy.

- The relevance of some between-species comparisons is unclear (e.g. herbivorous fish). Please explain how these relate to the carnivorous grouper model. When comparing to other species, explain the relevance to the current study.

- The discussion jumps between topics in places. A more linear flow from growth to FCR to morphology would improve clarity. 

Some long, complex sentences could be shortened or split into two sentences for improved readability. 

- Watch for repetitive words/phrases. For instance, "trend" is used frequently - consider substituting synonyms like "pattern" or "change."

Author Response

Reply to the Review Report

Reviewer 2

Comments and Suggestions for Authors

This ms examines the effects of dietary carbohydrate-to-lipid ratio (CHO:L) on growth performance, body composition, intestinal morphology, and glycolipid metabolism in juvenile hybrid grouper fish. The topic is relevant given the lack of specific nutritional guidelines for this emerging aquaculture species. The study design using 6 isonitrogenous, isoenergetic diets with graded CHO:L ratios is appropriate. The data appears to be thoroughly collected and analyzed. However, some issues need to be addressed:

Reply: Thank you very much for recognizing our work and giving us valuable comments. The manuscript has been carefully revised in accordance with your comments.

Abstract:

Line 20-21: Please clarify if you meant "specific growth rate" here.

Reply: Thank you very much for helping us point out the errors. Your suggestions have made our manuscript more rigorous. We have corrected line 21 of the manuscript according to your suggestion. In addition, we checked for similar errors in the full manuscript to ensure the quality of the manuscript.

Line 29-30: The statement about the quadratic regression model is redundant since it is mentioned earlier. Consider removing.

Reply: Your suggestion helped us to streamline our manuscript even more. Thank you very much for your valuable comments. Lines 28-29 of the manuscript have been revised as you suggested.

M&M:

Please add details on the feeding strategy during acclimatization.

Reply: Thank you very much for your valuable suggestions. Your suggestions have made our manuscript more detailed. In lines 102-107 of the manuscript, we have added the feeding strategy during acclimatization.

On lines 123 and 179, "protein efficiency rate" is used to describe the protein efficiency ratio (PER) calculation. PER is the standard term for this metric in the aquaculture nutrition literature. I suggest revising the text on lines 123 and 179 to use the terminology "protein efficiency ratio (PER)" rather than protein efficiency rate. This edit will align the terminology with the accepted standard in the field.

Reply: Thank you very much for your extremely careful help in pointing out the problems. Your suggestions have been a great help in improving the quality of our manuscript. In lines 121 and 183 of the manuscript, “protein efficiency rate” has been changed to “protein efficiency ratio”.

Expand section 2.4 to provide more information on the regression models and variables used in the analysis.

Reply: Thank you very much for your suggestion. We have added information about the statistical analysis in lines 161-165 of the manuscript as you suggested.

Discussion:

Consider explaining the potential mechanisms and biological significance of key findings. For example, the increase in glycolytic enzymes at intermediate CHO:L ratios likely reflects greater carbohydrate catabolism and energy production. This context would add helpful physiological insights.

Reply: Thank you very much for your suggestions. Your suggestions are very helpful in improving the quality of our manuscripts. In lines 354-359 of the manuscript, we address the biological significance of the enhanced glycolytic capacity under the appropriate CHO:L ratio. In lines 391-394 of the manuscript, we speculate or elaborate on the biological significance of fatty acid oxidation and fatty acid degradation.

Summarize the main points to improve flow and reduce redundancy.

Reply: Thank you so much for your patience in helping us improve the quality of our manuscript. We have reorganized the content of the manuscript to make it more focused and streamlined.

The relevance of some between-species comparisons is unclear (e.g. herbivorous fish). Please explain how these relate to the carnivorous grouper model. When comparing to other species, explain the relevance to the current study.

Reply: I appreciate your advice very much. There are differences in the optimum CHO:L ratios for different dietary fish. It was generally agreed that the optimum CHO:L ratio of diets for herbivorous fish should be the highest, followed by omnivorous fish and finally carnivorous fish. In lines 280-286 of the manuscript, we describe the similarity of the trend of our experimental results with the existing reported results for various predatory fish. The credibility of the results of this experiment is clarified. In lines 305-310 of the manuscript, we compare our experimental CHO:L ratios with those of herbivorous and omnivorous fish, and find that our experimental results are consistent with the generally accepted pattern.

The discussion jumps between topics in places. A more linear flow from growth to FCR to morphology would improve clarity.

Reply: Thank you very much for your suggestions. In order to make the manuscript more holistic, we have run through the indicators during the discussion. It did cause inconvenience to the readers in the process of reading. We have reordered the discussion section as you suggested.

Comments on the Quality of English Language

Some long, complex sentences could be shortened or split into two sentences for improved readability. Watch for repetitive words/phrases. For instance, “trend” is used frequently - consider substituting synonyms like “pattern” or “change”.

Reply: Thank you very much for your suggestions. We have taken apart and revised the long difficult sentences in the manuscript. Replacement of recurring words was done. In addition, our manuscript was submitted to a native English-speaking expert for linguistic adjustments and revisions.

Reviewer 3 Report (New Reviewer)

The paper is good, I would like to see it published. However, there is confusion regarding statistics. In the material and methods it does not mention the polynomial regressions that are in the results. The results must contain only the correct statistic and not various information as it is. Example: If the result was linear, the table does not need to have letters indicating differences. I indicate mandatory corrections and complete cleaning of the tables with correct and pertinent information.

Author Response

Reply to the Review Report

Reviewer 3

Comments and Suggestions for Authors

The paper is good, I would like to see it published. However, there is confusion regarding statistics. In the material and methods, it does not mention the polynomial regressions that are in the results. The results must contain only the correct statistic and not various information as it is. Example: If the result was linear, the table does not need to have letters indicating differences. I indicate mandatory corrections and complete cleaning of the tables with correct and pertinent information.

Reply: Thank you so much for recognizing our manuscript. Your suggestions have made our manuscript more rigorous. We have added information on polynomial regression analysis in the “Materials and Methods” section as you suggested (Lines 161-165). If the p-value of the treatment is less than 0.05, this indicator is considered to be significantly different. The letters in the table are used to distinguish the differences in the values among the treatments. The p-values of the linear is simply a response to the trend of the metrics across treatments. If the p-value of linear is less than 0.05, the use of letters to indicate differences can present a very accurate picture of the specific differences among the different treatments. In most of the reports, it is in this form that the differences in the data are represented in more detail.

For example:

Hougaina Panmei, Prasanta Jana, Tincy Varghese, Paul Nathaniel T, Narinder Kumar Chadha, Gopal Krishna, Gour Hari Pailan, Subrata Dasgupta, Dietary magnesium chelate alleviates oxidative stress and improves growth in white-leg shrimp, Penaeus vannamei (Boone, 1931), reared in inland saline water. Animal Feed Science and Technology, 2023, 303: 115692. https://doi.org/10.1016/j.anifeedsci.2023.115692.

G.M. Siddaiah, Rajesh Kumar, Rakhi Kumari, N.K. Chandan, Jackson Debbarma, D.K. Damle, Arabinda Das, S.S. Giri, Dietary fishmeal replacement with Hermetia illucens (Black soldier fly, BSF) larvae meal affected production performance, whole body composition, antioxidant status, and health of snakehead (Channa striata) juveniles. Animal Feed Science and Technology, 2023, 297: 115597. https://doi.org/10.1016/j.anifeedsci.2023.115597.

Xuhui Zhang, Zhiyuan Sun, Jinfeng Cai, Guibin Wang, Jiahong Wang, Zunling Zhu, Fuliang Cao, Dietary supplementation with fermented moringa oleifera leaves inhibits the lipogenesis in the liver of meat ducks. Animal Feed Science and Technology. 2020, 260: 114336. https://doi.org/10.1016/j.anifeedsci.2019.114336.

Round 2

Reviewer 3 Report (New Reviewer)

Satisfied with the improvements

This manuscript is a resubmission of an earlier submission. The following is a list of the peer review reports and author responses from that submission.

Round 1

Reviewer 1 Report

Abstrac

My principal concern is that the abstract does not describe the problem Why? And the authors don’t have a clear hypothesis, in consequence, they don’t have a clear implication of their results. This section describes methods and result…………

Introduction

The research problem is no defined,

Which topic is addressed in this research, Nutrition per se? strength of a hybrid fish? CHO diet use?

Where is the hypothesis of the research?

Which is the Gap missing in this topic?

Methodology

Line 102-110: be more specific in the culture environment description, Recirculating aquaculture system? Open system? Photoperiod? How were the variables controlled? Please describe,

Line 112; why did you kill 30? Was necessary?

Line 114: You were breeding the fish?

Line 122: In total # of fish, how many fish do you use for sampling from each tank?

 The entire 2.3 section is so confusing in the number o fish that the author uses during the experiment, please clarified,

Section 2.4: why are you using data published with anteriority in this paper? Variance test? Normality test? Parametrical or no parametrical data distribution?

Result

The result description not match with the methodology of this research, please be sure the all results have been previously described in methodology section

Fig 2; description?

Fig 3; description?

Discussion

Need to be improved after major revision

Reviewer 2 Report

Examination of the nutrition of farmed fish, including the effects of feed formulations on the growth, serum biochemical index, intestinal morphology and enzyme activity for digestion, ability to metabolize carbohydrate and lipid in hybrid grouper (Epinephelus fuscoguttatus ♀ × E. lanceolatus ♂), is appropriate for the journal. The authors have collected quite a lot of information, but there are some shortcomings with data analyses. There are weaknesses in the presentation that can be corrected.

It should be emphasized that the fish in the experiment are very young, with high growth potential, where FCR has no significant consequences on the final production costs and where FCR is one of the indicators for the nutritional efficiency of the offered food formulations.

In order to be as clear as possible in revising this paper, I will start from the conclusion (lines 387-389) (Under the conditions of this experiment, based on SGR of pearl gentian grouper, after 387 fitting by the second-order polynomial regression model, the best dietary CHO:L ratio 388 was 1.72.) which is factually maybe correct, but may lead to a misinterpretation of the obtained result. That is, first the SGR is listed among the indicators that do not differ significantly according to the criteria of different food formulations, and then, based on the polynomial regression analysis of the relationship between food composition and SGR, the authors conclude that certain food formulations give a higher SGR. Such a possible discrepancy should be well explained or the conclusion cannot be unambiguously written as it is in the manuscript. Moreover, the same could be concluded from the regression analysis of the relationship between CF and food formulation (which was not done), so it is not clear whether the SGR is only under the influence of a greater amount of accumulated fat and water in the body (Table 3.) or can we really talk about growth? The thesis about the influence on growth could be strengthened if the influence of food composition on length growth was shown, but as I understood from the materials and methods, length was not measured in a sufficient number of fish to be able to do this analysis.

If the influence of food composition on growth cannot be unambiguously determined, the conclusions should be focused on the results of the remaining research results.

The claim in the material and methods that foods are isoenergetic (line 84) is not consistent with the display of gross energy (MJ/Kg) in Table 1, which needs to be clarified. It is also necessary to take into account that the increase in the fat content in the prepared foods is mainly due to the addition of vegetable fats, while the proportion of fish oil was constant. What if it was the opposite? if there is a possibility that the result would not be equal, then the discussion cannot be reduced to the total amount of fat.

Reviewer 3 Report

This manuscript assesses the effect of diets with varying CHO and lipid ratios and equal protein and energy for hybrid grouper. The study presents a comprehensive dataset of parameters including growth, histology, enzyme activity, serum biochemistry relating to lipid metabolism. The presentation and discussion of the manuscript requires attention. Spelling and grammar mistakes are present throughout. Further explanation of the justification of dietary formulation is required. The discussion should relate back to the concepts of protein-sparing, energy preferences and fat deposition that were highlighted in the introduction. Connection between the many parameters assessed and these broader concepts are required to understand how nutritionists can use the information presented to formulate better feeds for hybrid grouper. This cannot be done at present because the manuscript also presents references with contrasting results and does not provide clarity on how to interpret the data from this study compared to the results of others.

Further comments include

Line 58. What is the optimal dietary lipid for this species? Has lipid requirement studies been completed? What about for protein, energy and CHO (thresholds)? Provide evidence for the decision to formulate the diets at 50% protein and vary the range of lipid from 12 to 18.8%?

Line 19. ‘not’ instead of ‘non’

Line 45. ‘consuming’ instead of ‘dining’

Line 57. “dietary lipid’ instead of ‘feed lipid’

Lne 75. ‘profile’ instead of ‘patterns’

Line 80. ‘digestive enzyme activities, and carbohydrate and lipid metabolism’ instead of ‘enzyme activities of digestive, carbohydrate and lipid metabolism’

Line 89. Mention the ingredients which contributed to the CHO level (corn starch and cellulose), based on a formulated target of dietary CHO. This was not confirmed with CHO analysis but can calculate CHO amount based on the difference of ash, protein, lipid composition?

Line 107. “Feed was delivered twice... 16:00 to apparent…”

Line 108. “the feeding trial, rearing conditions were maintained so that…”

Line 120. ‘to determine’

Line 12. Incomplete sentence. Please proof-red the manuscript as there are many examples of this.

Table 3. It would be more meaningful to present the dry matter composition rather than the moisture composition.

Line 175. Sentence is too long with too many subjects. Break sentence up and talk about each finding or comparison in a new sentence.

Line 277. There are contradicting sentences here. Similar findings are listed base on having similar results to the study but this it is suggested that grouper have a stronger ability to use carbohydrate. Provide clarity.

Line 286. This paragraph is confusing. The the SGR regression showed the optimal CHO:L is 1.72. Another study revealed the optimal CHO:L was 0.77 but the grammar of the sentence is confusing and suggests that it is referring to the current study. Please provide more detail of the other study mentioned. Based on which parameter?

It would be useful to understand the optimal CHO:L based on the other parameters which had a significantly difficult result (EE body composition, EE in liver/muscle, lipase, intestinal morphology, blood serum, liver enzyme activity). Are these parameters in agreement with the SGR result and are they correlated?

The objective of the study was to understand the ideal CHO:L to spare protein from being catabolised as energy. Provide more discussion around this aspect rather than focusing so much on the parameters and how they relate to lipid metabolism. How do these parameters relate to the bigger picture of efficiency energy and the undesirable deposition of too much body fat?

The quality of english requires improvement and would provide more clarity of the author's meaning.